Influence of phragmites density, algal concentration and water velocity on cyanobacterial bloom dynamics

Lv Jiaming 1 2
Yang Guijun 2
Zhang Yuqing 3
Shao Keqiang 1 4
Tang Xiangming xmtang@niglas.ac.cn 1 4
1 State Key Laboratory of Lakes and Environment, Nanjing Institute of Geography and Limnology, Chinese Academy of Sciences , Nanjing , China
2 College of Environment and Ecology, Jiangnan University , Wuxi , China
3 The Third Construction Company of CCCC Second Harbor Engineering Co., Ltd. , Zhenjiang , China
4 College of Resources and Environment, University of Chinese Academy of Sciences , Beijing , China
Wang Liang
Electronic publication date: 2025 Jul 16
Publication date: 2025
Volume: 13
Electronic Location ID: e19704
Received 2024 Aug 6; Accepted 2025 Jun 13
Copyright: ©2025 Lv et al.
Copyright year: 2025
Copyright holder: Lv et al.
License: This is an open access article distributed under the terms of the Creative Commons Attribution License, which permits unrestricted use, distribution, reproduction and adaptation in any medium and for any purpose provided that it is properly attributed. For attribution, the original author(s), title, publication source (PeerJ) and either DOI or URL of the article must be cited.
License URL: https://creativecommons.org/licenses/by/4.0/

Keywords: Lake Taihu, Cyanobacterial blooms, Bacterial diversity, Phragmites wetland, Water velocity

Funding: Natural Science Foundation of Jiangsu Province, China BK20220018 National Natural Science Foundation of China 41971062 Taihu Light Scientific and Technological Research M20221002 This work was supported by the Natural Science Foundation of Jiangsu Province, China (BK20220018), the National Natural Science Foundation of China (grant number: 41971062), the Taihu Light Scientific and Technological Research (No. M20221002). The funders had no role in study design, data collection and analysis, decision to publish, or preparation of the manuscript.

==============================
Background

Cyanobacterial blooms present a significant global water challenge, often accumulating in lakeside wetlands and impacting water quality. Despite this, wetland characteristics influencing bacterial diversity during cyanobacterial bloom degradation remain unclear.

Methods

To address this gap, we conducted a 30-day simulation experiment near Lake Taihu, China, to investigate the effects of Phragmites density, algae concentration and water velocity on bacterial diversity and water quality. An orthogonal design with three factors and levels was used with 18 tanks, each with a soil layer. Phragmites density, algae concentration and water velocity were adjusted to simulate lake conditions. Physicochemical parameters were measured within a month, and water samples were collected for bacterial biomass and DNA extraction. Bacterial 16S rRNA gene sequencing was performed to assess diversity, and statistical analyses including α-diversity, β-diversity, and analysis of similarities (ANOSIM) were conducted to evaluate the impact of the experimental factors on water quality and bacterial community structures.

Results

Algal concentration and water velocity had a greater impact on water quality than Phragmites density. Employing 16S rRNA gene sequencing technology, we discovered that bacterial α-diversity was significantly affected by phragmites density, water velocity, and time (P < 0.01), whereas bacterial β-diversity was significantly influenced by algal concentration and time (P < 0.001). The bacterial community structure was significantly impacted by phragmites density, water velocity, algal concentration, and time (P < 0.001). During the degradation of cyanobacterial blooms, the most abundant bacteria were Proteobacteria (36.8%), Bacteroidetes (20.4%), Cyanobacteria (19.1%), and Actinobacteria (10.3%). Algal density had a stronger influence on bacterial community structure than Phragmites density or water velocity. Orthogonal test results indicated that high algal concentration, coupled with reduced Phragmites density and increased water velocity, rapidly decreased nitrogen, phosphorus concentrations, and bacterial diversity. These findings deepen our understanding of Phragmites wetland effects on cyanobacterial blooms, offering insights for water ecological conservation and resource management in cyanobacteria-affected lakes.

Introduction

Wetlands are complex ecosystems found in the transitional zones between water and land (Xiao et al., 2022), they are of great importance for human survival and environmentally balanced socio-ecological progress. However, the massive development of cyanobacteria can degrade ecosystems and water quality in eutrophic lakes (Sun et al., 2016). Lakeside Phragmites wetlands are one of the most affected areas where cyanobacteria accumulate (Wu et al., 2022). It has been observed that the restoration of Phragmites wetlands along the lakeshore has resulted in the occurrence of cyanobacteria blooms in the water, which subsequently enter the Phragmites wetlands due to hydrodynamic force (Xin et al., 2020). These cyanobacteria blooms are then blocked by the Phragmites in the wetland, leading to their gradual accumulation, settling, decomposition, and sometimes even the formation of small-scale black water bloom (Zhou et al., 2021). Water movement spreads Cyanobacteria to surrounding areas, significantly impacting the local aquatic environment (Zhu et al., 2020). The accumulation and sedimentation of these blooms in lakeside Phragmites wetlands is altering the trophic dynamics of littoral ecosystems.

Bacteria play a crucial role in material circulation and energy flow within lake ecosystems, contributing significantly to maintaining ecosystem balance and driving elemental cycling (Cole et al., 2000; Newton et al., 2011). By examining the spatial and temporal distribution patterns of bacterial diversity and community structure in lakes, we can uncover their adaptation mechanisms to environmental changes and predict how lake ecosystems will respond to such changes. It is well-established that in aquatic ecosystems, the growth of algae stimulates the growth of bacteria (Qian & Xu, 2002). Symbiotic network analyses indicate that the relationships between algae and bacteria are predominantly positive, suggesting a mutualistic relationship (Yan et al., 2021). Limited research has directly investigated how Phragmites influences bacterial communities in the water column. Changes to Phragmites wetland conditions can indirectly affect soil properties, which in turn can impact the structure of bacterial communities within the wetland (Wang et al., 2021a). Apart from algae and Phragmites, hydraulic conditions also have an impact on the diversity and community structure of bacteria dwelling in the water column. A previous study indicated that flow disturbance intensities of 0.05 m/s and 0.10 m/s significantly enhanced the growth of phytoplankton and bacterioplankton (Cai et al., 2022).

To date, research on eutrophic lakes has primarily focused on the effects of cyanobacterial blooms on bacteria (Du et al., 2022; Huang et al., 2022). However, the degradation process of cyanobacterial blooms in lakeside wetlands, especially how environmental factors influence bacteria-mediated degradation of cyanobacteria and its environmental effects, still lacks in-depth research. The lack of quantitative data and analyses has hindered efforts to understand the detailed microbial mechanisms in eutrophic lakeside wetlands with persistent cyanobacterial blooms. To address the knowledge gap mentioned above, we conducted a 30-day microcosm orthogonal experiment with different levels of Phragmites density, algal concentration, and water velocity. The specific objectives of this study were to explore: (1) How does water quality change during the degradation of cyanobacterial blooms at different concentrations in lakeside wetlands? What are the primary influencing factors? (2) How do bacterial diversity and community structure respond during the degradation of cyanobacterial blooms? Which factor(s) plays a dominant role?

Materials and Methods

Experimental set-up and procedures

From September 19 to October 20, 2022, we conducted a 30-day simulation experiment on the shore of the eutrophic Lake Taihu, China. Employing an orthogonal design with three factors and three levels (Fig. 1 and Table 1), we utilized eighteen polyethylene water tanks (100 cm × 60 cm × 60 cm), divided into two groups with a 20 cm-deep soil layer at the bottom. Phragmites were planted at varying densities: approximately 50 plants/m2, 100 plants/m2, and 200 plants/m2, with the highest density reflecting the natural Phragmites density along Lake Taihu’s shore, as previously investigated in the northern lake (Lv, Shao & Tang, 2024).

Figure 1 The experimental setup comprised three vertically arranged treatment groups: Group 8 (top layer), Group 4 (middle layer), and Group 3 (bottom layer), with physical partitions separating each test section.

After allowing Phragmites to mature for three months, we introduced different algal concentrations and used various power pumps to simulate water flow. High-concentration algae water, collected from the Phragmites wetland along Lake Taihu’s shore, was proportionally diluted into three concentrations at the experimental site: approximately 600 ± 200 µg/L, 1,200 ± 200 µg/L, and 2,400 ± 200 µg/L, using BBE AlgaeLabAnalyser (Germany). The highest concentration mirrored the average algae concentration in the water along Lake Taihu’s shore (Zhu et al., 2020). Submerged water pumps operated at different settings to create intermittent water disturbances, simulating lake currents. Water flow velocities were set at approximately 0 m/s, 0.03 m/s and 0.1 m/s, with the maximum velocity resembling the water flow velocity in the lakeshore zone of Lake Taihu (Sheng et al., 2021). All tanks were positioned in the lakeshore area of the Taihu Laboratory for Lake Ecosystem Research (31°24′-N, 120°13′-E), part of the Chinese Ecosystem Research Network.

Table 1 Orthogonal test set-up.

We conducted a 30-day simulation experiment, employed an orthogonal design with three factors and three levels, and utilized total eighteen polyethylene water tanks in two replicates. Figure 1 shows the scene of the experiment.

Orthogonal Test Table	
Group	Phragmites density	Algal concentration	Water velocity	
	(plants/m2 )	(µg/L)	(m/s)	
1	low (50)	low (600)	low (0)	
2	low (50)	medium (1,200)	medium (0.03)	
3	low (50)	high (2,400)	high (0.1)	
4	medium (100)	low (600)	medium (0.03)	
5	medium (100)	medium (1,200)	low (0)	
6	medium (100)	high (2,400)	high (0.1)	
7	high (200)	low (600)	high (0.1)	
8	high (200)	medium (1,200)	medium (0.03)	
9	high (200)	high (2,400)	low (0)	

Sampling, physicochemical analyses and bacterial biomass collection

To minimize the influence of diurnal physiological variations, all samples were collected in the morning. Surface water samples (500 mL) were collected daily on days 0, 1, 4, 10, 20, 30 from each tank’s sampling area, totaling 108 water samples. On-site measurements included water temperature (WT), dissolved oxygen (DO), pH, and electrical conductivity (EC) using a YSI 6600 Multi-Parameter Water Quality Sonde. Standard methods (Jin & Tu, 1990) were employed for measuring total nitrogen (TN), total dissolved nitrogen (TDN), ammonium (NH3-N), total phosphorus (TP), total dissolved phosphorus (TDP), and chlorophyll a (Chl-a).

To quantify suspended solids (SS) (i.e., organic matter) in water samples, a 20–80 mL volume of surface water was filtered onto GF/F filters (Whatman, UK). Subsequently, the filters were dried in a drying oven for 4 h at 105 °C. Following the drying process, the GF/F filters were weighed using a ten-thousandth balance (Shimadzu Corporation, Kyoto, Japan). Then, the difference between the dried and pre-filtered weight was recorded as SS concentration. Next, the filters were then baked in a muffle oven at 550 °C for two hours. After baking, the filters were weighed again, and the difference between the weight after drying and after baking was LOI (loss on ignition). For bacterial DNA samples intended for diversity analysis, a 20–50 mL volume of surface water was filtered on 0.22 µm filters (Millipore Boston, MA, USA). The filters were then stored in sterile centrifuge tubes at −80 °C until DNA extraction.

Amplicon sequencing and analysis

All samples’ bacterial DNA was extracted using the FastDNA® Spin Kit for Soil (MP Biomedicals, Santa Ana, CA, USA) following the manufacturer’s protocols. The V3–V4 region of 16S rRNA gene was amplified by PCR using the bacterial universal primers 338F (5′-ACTCCTACGGGAGGCAGCAG-3′) and 806R (5′-GGACTACHVGGGTWTCTAAT-3′) (Fadrosh et al., 2014). The PCR was performed as previously described (Zhang et al., 2021). Purified amplicons were pair-end sequenced (2 × 300 bp) on the MiSeq platform (Illumina, San Diego, CA, USA) at Gene Denovo Biotech Co., Ltd. (Guangzhou, China). The original sequences obtained in this study have been submitted to the National Genomics Data Center (NGDC) of the Chinese National Center for Bioinformation (CNCB) with accession number CRA017920.

Bioinformatic analyses, including trimming the primer sequences, removing chimeras, clustering of operational taxonomic units (OTUs), filtering low abundance OTUs (<10 reads), and taxonomic assignment based on SILVA database 138, were conducted in the CLC Genomics Workbench 20.0 (Qiagen, Hilden, Germany). The operable taxon (OTU) clustering was performed with 97% sequence similarity. These procedures were performed following the tutorial and adhering to previous description (Xie et al., 2020).

Statistical analysis

Bacterial α-diversity, including richness, Chao1 and ACE, was calculated for different groups using a normalized sequencing depth of 6,547 reads with the R package “vegan” (Liu et al., 2023). Subsequently, Kruskal-Wallis tests were employed to assess significant differences in both time and group (Li et al., 2021c). For a more in-depth exploration of how different groups and times influence bacterial α-diversity, a one-way analysis of variance was conducted, and the results were visualized using Origin 2022 software.

To evaluate β-diversity, reflecting the variation of community structures among different groups and times, cluster analysis and non-metric multidimensional scaling (NMDS) were performed. These analyses utilized Bray–Curtis similarity based on the distance matrix of the bacterial community (Liu et al., 2023). Additionally, analysis of similarity (ANOSIM) was performed to test for bacterial compositional differences (Shen et al., 2022). The R value obtained from ANOSIM analysis ranges from 0 to 1, with larger R values indicating greater differences. The impacts of three factors on water quality and bacterial diversity were analyzed through orthogonal tests, with detailed information available in the Supplemental Information.

Results

Changes in physicochemical parameters

The variations in the main physicochemical parameters during the study are depicted in Fig. 2 and detailed in Table S1. Notably, the concentrations of TN, TP, and organic matter generally decreased, while the concentration of NH3-N increased throughout the experiment. The water temperature at sampling on days 0, 1, 4, 10, 20, and 30 recorded values of 25.6 °C, 23.9 °C, 24.9 °C, 27.3 °C, 19.5 °C and 20.4 °C, respectively. The concentration changes of TN, TP, OM, and NH3-N exhibited percentage reductions of −77%, −81%, −80%, and 81%, respectively, within the 30-day simulation experiment. In general, the TN concentration displayed a decreasing trend. Under high Phragmites density, low water velocity, and high algal concentration, TN levels increased noticeably between days 4 and 10 before significantly declining. Similar dynamics were observed in TP under conditions of high Phragmites density, high water velocity, and low algal concentration, while organic matter changes occurred within days 1–10. NH3-N exhibited an overall increasing trend.

Figure 2 The percentage concentrations of TN, TP, OM and NH 3 -N during the 30-day algae decomposition.

Data are means from the two times.

Upon analyzing the results of the orthogonal experiment (Table S3A–S3D), it was determined that algal concentration had the greatest influence on all water quality indices compared to the other two conditions. The order of influence of the three factors on the overall water quality index is as follows: algal concentration > water velocity > Phragmites density.

When Phragmites density is low, and algal concentration and water velocity are high and low, respectively, TN concentration decreases significantly (P < 0.05). Under the same conditions of low Phragmites density, high algal concentration, and high-water velocity, TP and OM exhibited a significant decrease (P < 0.05). Meanwhile, NH3-N showed a significant increase (P < 0.05) when all factors were at a high degree.

Dynamics of bacterial α-diversity

Bacterial α-diversity varied significantly across groups and time points (P < 0.05), as shown in Fig. 3. The average Chao1 index for all samples was 1,188 (Table S2). Groups 1, 2, 3, Groups 4, 5, 6, and Groups 7, 8, 9 exhibited similar patterns of change, with Groups 1, 4, 7 having the highest and Groups 3, 6, 9 having the lowest diversity. Throughout the experiment period, the Chao1 index showed a trend of fluctuation decline, and the variation was generally significant (P < 0.05). In the initial 10 days of the experiment, the Chao1 index decreased rapidly. However, after day 10, there was a slight rebound process.

Figure 3 Bacterial α-diversity across different groups and times.

Diversity index, as represented by Chao1, was calculated using a normalized sequencing depth of 6,547 reads.

Bacterial α-diversity was significantly influenced by three factors and time, and their interactions also had an impact. The orthogonal experimental results revealed that water velocity had the greatest influence on the Chao1 index compared to the other two conditions, followed by algal concentration (refer to Table S3E for details). The Chao1 index showed a significant decrease (P < 0.05) under conditions of high Phragmites density, high algal concentration, and high-water velocity.

Variation of bacterial community structures and compositions

Bacterial β-diversity showed distinct response patterns across groups and time points (Fig. 4). The Bray–Curtis dissimilarity index showed no significant differences in bacterial communities between groups (P > 0.05), but significant temporal variations were observed (P < 0.001). ANOSIM results indicated that the structure of the bacterial community showed significant differences with different levels of three factors and increasing time (P < 0.001). The analysis of orthogonal experimental results (Table S3F) revealed that algal concentration had the greatest impact on β-diversity compared to the other two conditions. The order of influence of the three factors was algal concentration, water velocity, and Phragmites density. The β-diversity index increased most significantly (P < 0.05) under the condition of medium Phragmites density, low algal concentration and high-water velocity.

Figure 4 The β-diversity of the bacterial community structure exhibited varying response patterns across different groups and times.

Bacterial community variation and β-diversity patterns across groups and times. NMDS ordination and box plot of bacterial community based on the Bray–Curtis distance. Differences in community structure among groups and times were tested using ANOSIM and Kruskal–Wallis tests.

The composition of the bacterial community underwent significant changes across different groups and time periods, as shown in Fig. 5. During the 30-day decomposition of cyanobacterial blooms, the relative abundance of Proteobacteria peaked at 45.7% on day 4. The relative abundance of cyanobacteria reached its maximum of 26.3% at the beginning of the study, while the relative abundance of Bacteroidetes peaked at 24.7% on day 4. The top five bacterial communities had an average relative abundance of 36.8%, 20.4%, 19.1%, 10.3%, and 8.3%, respectively. The relative abundance of Proteobacteria, cyanobacteria, and Actinobacteria fluctuated with time. The relative abundance of Bacteroidetes and Verrucomicrobia remained stable from day 0 to 4, but increased rapidly after day 10. The results of the orthogonal analysis indicate that the concentration of algae remains the primary factor affecting the composition of the bacterial community, followed by the density of Phragmites (Table S3G).

Figure 5 The composition of the bacterial community underwent significant changes across different groups and time periods.

Variation trends of dominate bacteria with increasing time (A). Bacterial community structure variation across groups and times (B). The positive correlation between Bacteroidetes and nutrient during the 30-day algal decomposition (C).

Discussion

The influence of wetland factors on water quality

The findings unequivocally reveal that individual environmental factors, as well as their synergistic interactions, exert a substantial influence on wetland water quality, as illustrated in Fig. 2. Our data show a marked reduction in TN concentration under conditions characterized by low Phragmites density, high algal concentration, and low water velocity. In a similar vein, TP and OM concentrations were significantly reduced under the same conditions, albeit with an increase in water velocity. The hierarchy of impact exerted by the three principal wetland variables on the water quality parameters assessed in our study remains consistent: algal concentration > water velocity > Phragmites density.

Chl-α concentration and algal density are critical indicators for assessing water quality in Lake Taihu (Lyu et al., 2020). Our results demonstrate that aquatic phosphorus concentrations are co-regulated by algal biomass and Phragmites biomass, with algal accumulation showing particularly strong associations with internal phosphorus loading (Wang et al., 2022). Over the 30-day study period, we observed that fluctuations and declines in algal concentration closely paralleled changes in TN and TP, as depicted in Fig. 2 and detailed in Table S1. In waters with a high trophic level index (TLI), the water quality index (WQI) was notably higher during the cyanobacterial decomposition phase compared to the growth phase. Conversely, in waters with a low TLI, the WQI was lower during the decomposition phase (Li et al., 2021b). Under high algal concentration conditions, groups 3, 6, and 9 were found to be rich in TN and OM (Redfield, 1958). These groups also initiated water environment conditions akin to hypoxia, as indicated in Table S1, leading to a gradual decline in ammonium nitrogen (NH3-N) during the early stages of the experiment, as shown in Fig. 2. Post-experiment Day 1, all water groups were found to be in an aerobic state, with the observed increase in NH3-N levels suggesting nitrification, the finding that aligns with previous studies (Liu et al., 2019; Shi et al., 2017). The varying release rates of carbon (C), nitrogen (N), and phosphorus (P) during aerobic decomposition are likely attributable to the initial C:N:P ratios of phytoplankton (Tezula, 1989). By the conclusion of the 30-day experiment, a 20% surplus of TN and TP remained in the water, potentially due to incomplete algal decomposition, underscoring that algal concentration remains the predominant factor affecting water quality.

Water disturbance (i.e., water velocity) has been shown to enhance the decomposition and mineralization of algae, as well as the release of phosphorus, with the intensity of the disturbance correlating positively with the magnitude of these effects, as depicted in Fig. 2. Nonetheless, when disturbance is excessive, it can trigger substantial sediment resuspension. This process allows suspended solids to adsorb dissolved substances from the water, consequently leading to a reduction in the concentration of dissolved reactive phosphorus within the aquatic environment (Zhang et al., 2022a). Throughout the experiment, the water maintained a high level of dissolved oxygen, likely a consequence of the disturbance. Under aerobic conditions, the release of phosphorus from sediments is reduced, which in turn limits the input of phosphorus into the water to a certain extent (Ni & Wang, 2015; Yu et al., 2023). Moreover, the release of nitrogen from sediments is largely suppressed in the absence of water disturbance (Wu et al., 2012), which is a primary pathway contributing to the overall reduction in nitrogen levels. Consequently, moderate disturbance during the decomposition of algae can help to mitigate the deterioration of water quality.

As anticipated, low Phragmites density expedited the dilution of algal concentrations, while high Phragmites density mitigated water velocity (Wang & Wang, 2008). Phragmites not only has the capacity to retain nitrogen and phosphorus but also possesses a more robust ability to purify these nutrients compared to other emergent plants, thereby reducing their concentrations in the water to a certain extent (Deng et al., 2013; Liu et al., 2012). However, the growth and development of algae are contingent upon the availability of nitrogen and phosphorus, which accounts for the observed decrease in their concentrations during the initial phase of the experiment (Xu et al., 2019). Consequently, Phragmites and algae engage in a competitive struggle for these nutrients (Li et al., 2021a). In the later stages of cyanobacteria decomposition, Phragmites decay and algal residues coexist in close proximity. Research indicates that the concurrent decomposition of algal and Phragmites debris exhibits a co-metabolic effect, which can hasten the release of nutrient salts and organic matter into the water (Shi et al., 2021). This process can exacerbate water quality degradation and intensify eutrophication.

The influence of wetland factors on bacterial community

Bacterial α-diversity varied significantly across different groups and times. Orthogonal analysis revealed that bacterial α-diversity is predominantly influenced by water velocity, with algal concentration being the next most significant factor. Notably, the Chao1 index exhibited a substantial decline under conditions of high wetland conditions. The fluctuations in bacterial α-diversity and physicochemical parameters observed in our experiment (Figs. 2 and 3) align with previous research, which has demonstrated that bacterial diversity correlates with variations in water environmental factors (You et al., 2023b). Disturbances leading to sediment resuspension disperse bacteria into the water, altering nutrient concentrations and the phytoplankton community structure (Shao et al., 2014; You et al., 2023a). This is corroborated by the observed reduction in TP concentration under high water velocity in our study. The general decline in both nutrient levels and bacterial α-diversity underscores the pivotal role of N and P as limiting factors for bacterial community growth and their close association with algae (Tang et al., 2017). General linear model analysis indicated a significant positive correlation between salinity and the Chao1 index throughout the decomposition process. Prior studies have established that bacterial communities are markedly affected by salinity, which is regulated by nutrient salts in the water (Park et al., 2021; Zhang et al., 2022b). These effects are ultimately attributed to the changes in water disturbance and algal concentration under the experimental conditions.

The β-diversity of the bacterial community structure revealed distinct response patterns across different groups and time periods, as illustrated in Fig. 4A. Orthogonal analysis indicated that bacterial β-diversity was predominantly influenced by algal concentration, with water velocity being the secondary factor. Notably, there was a significant increase in β-diversity under conditions of medium Phragmites density, low algal concentration, and high water velocity. Although no significant differences in β-diversity were observed between groups, it did increase significantly over time, coinciding with a general decline in nutrient concentrations throughout the experiment, as shown in Figs. 2 and 4B. Our study detected an inverse relationship between the bacterial community and algal biomass, which is in line with previous research (Li et al., 2022). On Day 4 and Day 30 of the experiment, the average algal concentrations in the water were 777 µg/L and 106 µg/L, respectively. The community structure diversity reached extreme and maximum values at these points, suggesting that algal blooms tend to diversify under these concentrations, which is an effective method for controlling algal blooms (Wang et al., 2021b). Adverse environmental conditions may foster the emergence of various habitat-specialized species, thereby enhancing the heterogeneity of bacterial communities (Shen, Juergens & Beier, 2018). Previous studies have demonstrated that frequent and chaotic hydrodynamic disturbances tend to homogenize and randomly aggregate the community structure of planktonic bacteria (Bai et al., 2020; Mo et al., 2018). However, the disturbance pattern in our experiment was intermittent and orderly, differing from the hydrodynamic conditions in Lake Taihu (Qin et al., 2023). The results suggest that the β-diversity in the group with high water velocity was not significantly higher than that in the group with low water velocity. Therefore, under these experimental conditions, algal concentration emerges as the primary factor affecting β-diversity, rather than water disturbance.

Algal concentration emerged as the primary determinant of bacterial community structure, with Phragmites density being a secondary factor. Upon the introduction of fresh algae into the microcosm, the initial dominant bacterial taxa identified were Proteobacteria, cyanobacteria, Bacteroidetes, and Actinobacteria, excluding Verrucomicrobia-groups commonly observed in Lake Taihu (Feng, Yuan & Wang, 2017; Niu et al., 2011; Qian et al., 2018; Xue et al., 2018). Throughout the 30-day experiment, Proteobacteria remained the predominant group, a pattern consistent with a 95-day study on bloom decomposition (Shi et al., 2017). The study revealed that the concentrations of TN and TP likely have a direct influence on the abundance of cyanobacteria, which declined over time (Figs. 2 and 5A). This suggests that cyanobacteria are sensitive to N and P, nutrients essential for bacterial growth and reproduction (Sheng et al., 2016). The initial anaerobic environment may impact the stability of the bacterial community (Xue et al., 2018; Zhou et al., 2022). On Day 4 of the experiment, Proteobacteria and Bacteroidetes experienced a significant increase, becoming the dominant species in the samples (Shi et al., 2017). Additionally, Bacteroidetes were observed to have a positive correlation with nutrient levels, in contrast to Proteobacteria (Fig. 5C) (Bai et al., 2023). Strains of Bacteroidetes play a crucial role during cyanobacterial blooms, as they are capable of degrading both macromolecular compounds and microcystis cells (Cottrell & Kirchman, 2000; Eiler & Bertilsson, 2004; Yamamoto et al., 1993). Bacterial diversity began to increase on Day 10, coinciding with the rapid decomposition of algal blooms and a swift increase in inorganic nutrients (Figs. 2 and 5A). This led to a significant rise in the relative abundance of Actinobacteria and Verrucomicrobia, indicating a decline in water quality. This is in agreement with the findings of a 14-day mesocosm experiment during the decomposition of microcystis blooms (Shao et al., 2014). Actinobacteria and Verrucomicrobia strains were found in high abundance during cyanobacterial blooms (Berg et al., 2009; Pope & Patel, 2008) and became predominant at day 20. Furthermore, the loss of large plants like Phragmites can significantly reduce habitat heterogeneity, leading to a decrease in available niches within ecosystems and increased competition among species for limited resources (Shade, Jones & McMahon, 2008). However, in this experiment, the impact of Phragmites density on bacterial community structure was not significant. The most critical factor was the direct or indirect effect of nutrient conversion during the decomposition of algal blooms, a process that is ongoing.

Conclusion

In this study, we explored the interplay between water quality, bacterial diversity, and community structure within 18 mesocosm ecosystems, subjected to nine orthogonal combinations of Phragmites density, algal concentration, and water velocity over a 30-day period of cyanobacteria decomposition. Our quantitative simulation experiments revealed that nutrients in the water undergo transformations among various forms during cyanobacterial growth, leading to significant alterations in key physicochemical water parameters, including TN, TP, OM, and NH3-N. However, algal concentration was identified as the paramount factor influencing all these indicators. Bacterial α-diversity exhibited substantial variation across different groups and times. Under conditions of high algal concentration, reducing Phragmites density and increasing water velocity were found to significantly lower the concentrations of N, P, and OM in the water. Conversely, decreasing Phragmites density and water velocity significantly enhanced bacterial α-diversity within water bodies characterized by high algal concentration. Over the 30-day decomposition of algal blooms, bacterial β-diversity experienced a notable increase, with algal concentration emerging as the primary determinant of bacterial diversity.

Furthermore, the bacterial community structure responded dynamically to different groups and times, with Proteobacteria, Bacteroidetes, cyanobacteria, and Actinobacteria being the dominant phyla throughout the decomposition process. This study has unveiled the complex interrelationships among algal concentration, Phragmites density, and water flow velocity in the context of water quality changes in lakeside wetlands. It also sheds light on their effects on bacterial diversity and community structure, offering a theoretical foundation for the scientific management and mitigation of eutrophication in lakeside wetland ecosystems.

Supplemental Information

Supplemental Information 1 Time distribution of water quality change under different conditions

Our orthogonal study conducted 6 times over a one-month period, two repetitions were taken for each sample to acquire the average. Sal, DO, TN, TP, OM and NH3-N were chosen to represent water quality changes during cyanobacterial decomposition. S represents the overall standard deviation of each group of samples. It indicates the degree of influence that different levels of factors have on water quality indicators.

Supplemental Information 2 The Alpha-diversity index of each sample

Diversity indexes, including Richness, Chao1, ACE, Shannon, Simpson and Invsimpson, were calculated using a normalized sequencing depth of 6547 reads. Two repetitions were taken for each sample to acquire the average. S represents the overall standard deviation of each group of samples.

Supplemental Information 3 The effects of three factors on water quality and bacterial diversity during cyanobacteria degradation were examined by orthogonal analysis

K value is the sum of the results of three levels of factors and reflects the influence of different levels and that R value the influence of different factors. S represents the overall standard deviation of each group of samples. It indicates the degree of influence that different levels of factors have on water quality indicators.

We are grateful to Jingchen Xue, Meng Qu, and Dong Li for their assistance with the sample collection and laboratory measurements.

Additional Information and Declarations

Competing Interests

Author Contributions

Data Availability

Yuqing Zhang is employed by the third Construction Company of CCCC second Harbor Engineering Co., Ltd.

Jiaming Lv conceived and designed the experiments, performed the experiments, analyzed the data, prepared figures and/or tables, and approved the final draft.

Guijun Yang analyzed the data, prepared figures and/or tables, authored or reviewed drafts of the article, and approved the final draft.

Yuqing Zhang performed the experiments, authored or reviewed drafts of the article, and approved the final draft.

Keqiang Shao conceived and designed the experiments, performed the experiments, authored or reviewed drafts of the article, and approved the final draft.

Xiangming Tang conceived and designed the experiments, performed the experiments, analyzed the data, prepared figures and/or tables, authored or reviewed drafts of the article, and approved the final draft.

The following information was supplied regarding data availability:

The sequences are available at the National Genomics Data Center (NGDC) of the Chinese National Center for Bioinformation (CNCB): CRA017920.

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
