# Peer review of "Influence of phragmites density, algal concentration and water velocity on cyanobacterial bloom dynamics"

_PeerJ, doi:10.7717/peerj.19704_

## Round 0.1 · original submission · Major Revisions

Please revise the manuscript based on all the comments by the reviewers.

Reviewer 1 ·

Basic reporting

Congratulations to the authors for bringing up the research which would significantly input the scientific community. The article " Influence of phragmites density, algal concentration and water velocity on cyanobacterial bloom dynamics " is a well-structures and executed research. But, it would be more impactful if the following minor changes are incorporated-
1. Lines 29-32 in the abstract guidance has to be rephrased as the meaning of the sentence is not conveyed properly.
2. In line 73, a type error is found which spells "dowelled" instead of "dwelled"
3. Line 73, the entire sentence can be rephrased for better understanding

Experimental design

The experiments are well planned and executed. But, I have a few queries about the experiments which are-
1. Was all the experiments done in repetition?
2. Why there is the comparison between algal concentration, phragmite density and water velocity?
3. How was the water density measured?
4. Why wasn't there a metagenomic analysis to compare the bacterial diversity?

Validity of the findings

The validity of the findings is pretty good.

Reviewer 2 ·

Basic reporting

English needs major revising, especially in the Discussion.

Experimental design

No comment.

Validity of the findings

Some sentences in the discussion confused correlation with causation. Many limitations to the dataset were not mentioned in the discussion. I provided more details in the next section.

Additional comments

I’m a bit concerned about the lack of replication in the study, it looks like the average of two replicates was calculated a lot. My understanding is that this technically “works” as in the code/software runs but it is generally extremely ill-advised and should be interpreted with the utmost caution, which I think needs to be made clearer in the discussion.

I found the discussion to be the hardest section to follow with the writing. I would recommend having someone with excellent English skills check it over. I think the uses of past and present tense and whether they were referring to past studies or the current study were the hardest parts. If I were to review this again I would want to have seen almost every discussion paragraph edited for additional clarity. Some of the sentences were also repetitive. Repeated mentions of “algal concentration having the most impact” is one example that stands out. While a summary of the main findings of the results can be a good thing in the discussion, there was almost too much results in the discussion rather than interpretation and broader implications.

This may be due to my own ignorance, so I maybe apologize, but I did get confused about the relationship between mentioning algae and cyanobacteria in the study. Although cyanobacteria were formerly referred to as ‘blue-green algae’, they are technically bacteria. So when you say “algal concentration” is that referring to the cyanobacteria concentration? Please clarify.

Throughout: Phragmites is a genus & therefore should be capitalized and italicized.

L26: “revealing significant variations” sounds more like results than methods.

Abstract: There is spacing in between (P < 0.01) on line 30 and no spacing between (P<0.001) on line 32, keep it consistent.

L31: Why put a p value for the alpha diversity & not beta diversity results?

L32: What is meant by “wetland variables”? Need the specifics unless you define ‘wetland variables’ earlier

L33: ‘Cyanobacterial’ shouldn’t be capitalized here.

Introduction:

L52: “observed to cause cyanobacteria blooms in the water to enter the phragmites wetlands” not sure what this part is supposed to say, is “to enter” supposed to say “entering” instead?

L58: Can you clarify what you mean by they are making ecology more complicated? Nature is always complicated.

L67-68: This sentence is repetitive, I suggest rewording to: “suggesting a mutualistic relationship.”

L73: I’m not sure what “dowelled” is.

L73: Suggest changing “Previous study” to “A previous study”

L77: Suggest changing “researches” to “research”

L85: Suggest changing “objective” to “objectives” and updating the rest of the grammar of that sentence to plural since multiple objectives are listed.

L88: Suggest changing “factor” to “factor(s)”

Materials & methods

L97: remove the spaces before “/m2”

L104: how did you measure these concentrations?

Starting at line 114, was any of the water quality collections done with sterile materials or sterile technique? If not, doesn’t this mean your tubs were routinely exposed to external bacteria? This is definitely worth a mention in the discussion.

I’m confused about the “20~80” and “20~50” mL volumes in the water sampling section. First, I haven’t seen this formatting before, I suggest changing to “~20-80” and “~20-50” if that’s what was intended. Second, how can you compare the collected weight of a sample that came from 20 mL of water to another that came from 80 mL of water? How do you know the actual weight per mL of water is different or it’s just 4 times as much weight because you filtered more of the materials from the higher volume of water to begin with?

L141: suggest changing “accession numbers” to “accession number”

L144: If you clustered into OTUs, then you need to provide the clustering rate (e.g. 97%, 99%, 100%).

L155: strange spacing between ‘visualized’ and ‘using’ here

L178: I don’t see why this should be its own paragraph when the first sentence discusses results that still connect with the previous paragraph.

L236: Figure 1 is an experimental setup, it doesn’t show any evidence of anything, so remove the mention of Figure 1 here please.

L245-6: What do you mean the pollution was affected “more” – more than what?

L255-8: The grammar of this sentence is off & I don’t follow it – the sentence beginning with “The present result,” and ending with “Shi et al., 2017”.

Paragraph from line 264-275: I’m confused about the discussion of disturbance, do you mean some kind of rain event happened during your experiment or what? Or are you calling the cyanobacterial bloom a disturbance event? Please clarify.

L295: What does “high wetland conditions” mean

L306: I don’t think it’s typical to include p values and R2 values in the discussion, they belong in the results.

L318: “observed a negative impact” – it was a correlation, but this sentence implies causation.

L321: I have no idea what the difference between an extreme value and a maximum value would be.

L323: “conducive to the formation” – same comment as L318, this is implying a direct cause and effect and order of events.

L351: “became involved” – similar comment as above, way too active of an assertion for a correlation.

Conclusion paragraph: I would remove p values, they belong in results. I would also cut most of this text that was already mentioned in several other places in the discussion.

Figure 1’s caption is woefully inadequate. For instance, please clarify what “x2” in the image is referring to so that the reader doesn’t need to revisit the methods section. Also please clarify that the pictures are representatives of some of the groups, not all 9 of the groups, if I understood correctly.

·

Basic reporting

no comment

Experimental design

no comment

Validity of the findings

no comment

Additional comments

I enjoyed reading this study. It was well designed and nicely written. The results were clearly articulated, and the methods mostly supported the findings. I have only a few comments below.
Line 96: Please detail the soil properties, or where the soil was taken from. Was it all from one source. Soil could be a source of bacteria and may affect the end results if different soils were used.
Line 122-127: Please check the methods for weighing suspended sediments. Shouldn’t the first oven drying and weighing occur before the water filtering, then after filtering the second oven dry, then re-weigh filter papers, and then take the difference in weight?
Line 168: Where do you present the suspended sediment (SS) results?
Also, organic matter (OM) is not mentioned in methods section. Is SS a proxy here for OM?

---

## Round 0.2 · Major Revisions

Please revise the manuscript by following the reviewers' comments.

Reviewer 1 ·

Basic reporting

No comments

Experimental design

No comments

Validity of the findings

No comments

Additional comments

The authors have well addressed the previous comments and revised the manuscript. Just check for the grammatical errors in the manuscript.

Reviewer 2 ·

Basic reporting

The English and grammar has been vastly approved since the last submission, but now I am confused by some of the content which requires additional clarification.

Experimental design

No comment

Validity of the findings

Some parts need clarification, I pointed them out in the attached document.

Additional comments

Attached in separate document.

Annotated reviews are not available for download in order to protect the identity of reviewers who chose to remain anonymous.

·

Basic reporting

Even though the manuscript presents a clear scientific narrative, running it through an AI application for grammatical corrections would increase readability, remove redundancy, increase conciseness and enhance clarity. For example:
First Paragraph
Original: "The movement of water causes it to spread to the surrounding areas, resulting in significant impacts on the water environment in those areas."
Revised: "Water movement spreads cyanobacteria to surrounding areas, significantly impacting the local aquatic environment."

Second Paragraph
Original: "Few studies have directly examined the effects of Phragmites on bacteria in the water column."
Revised: "Limited research has directly investigated how Phragmites influences bacterial communities in the water column."

Experimental Setup
Original: "After three months of Phragmites maturation, we introduced different concentrations of algae water and employed various power pumps."
Revised: "After allowing Phragmites to mature for three months, we introduced different algal concentrations and used various power pumps to simulate water flow."

Original: "Water pumps, placed below the water surface, featured different gears to simulate intermittent water disturbance mimicking lake currents."
Revised: "Submerged water pumps operated at different settings to create intermittent water disturbances, simulating lake currents."

Sampling Methods
Original: "To mitigate diurnal variations in cell physiology, all samples were collected in the morning."
Revised: "To minimize the influence of diurnal physiological variations, all samples were collected in the morning."

Original: "Then, the difference between the weight after drying and before filtering was SS as the result."
Revised: "The difference between the dried and pre-filtered weight was recorded as the suspended solids (SS) concentration."

Original: "Next, the weighed filters underwent a baking step in a muffle oven for 2 hours at 550℃."
Revised: "The filters were then baked in a muffle oven at 550℃ for two hours."

Statistical Analysis
Original: "Bacterial α-diversity exhibited significant variations (P < 0.05) across different groups and times, as illustrated in Fig. 3."
Revised: "Bacterial α-diversity varied significantly across groups and time points (P < 0.05), as shown in Fig. 3."

Original: "According to the Bray-Curtis non-similarity index, the bacterial community did not exhibit significant differences between groups (P > 0.05), but significant differences were observed with increasing time (P < 0.001)."
Revised: "The Bray-Curtis dissimilarity index showed no significant differences in bacterial communities between groups (P > 0.05), but significant temporal variations were observed (P < 0.001)."

Results and Discussion
Original: "However, under conditions of high Phragmites density, low water velocity, and high algal concentration, there was a noticeable increase during days 4-10, followed by a significant decrease."
Revised: "Under high Phragmites density, low water velocity, and high algal concentration, TN levels increased noticeably between days 4 and 10 before significantly declining."

Original: "The β-diversity of the bacterial community structure exhibited varying response patterns across different groups and times (Figure 4)."
Revised: "Bacterial β-diversity showed distinct response patterns across groups and time points (Fig. 4)."

Note that this is an example of AI grammar suggestions on only part of the manuscript. I highly recommend employing these methods for improving your communications to the scientific community.

Experimental design

no comment

Validity of the findings

no comment

Additional comments

Thank you for the new and improved manuscript.

---

## Round 0.3 · accepted · Accept

Please follow the reviewer's comments and thoroughly check the grammar and language use of the manuscript. After that, the manuscript can be accepted for publication by the journal.

Reviewer 1 ·

Basic reporting

The authors have well addressed my comments from previous revision. But there are spacing errors throughout the document. For eg, it's written as "500mL" rather than "500 mL". Please fix it and go for a thorough grammar check.

Experimental design

No comment

Validity of the findings

No comment